# ProofAug+: Boosting Reinforcement Learning for LLM Theorem Provers with Conditioned Proof Repair

## Abstract

Reinforcement Learning with Verifiable Rewards (RLVR) often suffers from the scarcity of positive samples on challenging tasks such as formal theorem proving. In this work, we propose ProofAug+, an RL training pipeline for LLM theorem provers that improves the training performance by acquiring more positive samples during rollout through ProofAug, a previously developed inference-time proof repair technique. The design of ProofAug+ is guided by two principles, progress guarantee and variance reduction, to address the performance degradation and policy collapse issues observed when integrating ProofAug into GRPO via naive direct replacement. These principles first lead to a novel LLM RLVR algorithm, Proximal Language Modeling Policy Optimization (PLPO), where in each iteration we use the exact objective as the optimization target instead of surrogate objectives used in TRPO/PPO and employ a gradient rejection mechanism to suppress large policy updates. Then, we integrate ProofAug into PLPO in a constrained way to achieve a balance between the exploitation of additional positive reward signals and the suppression of distribution shift that could violate the progress guarantee principle. Experiments show that PLPO achieves better stability than baseline GRPO-like algorithms while maintaining higher entropy during training. Building on PLPO, the resulting ProofAug+ pipeline further yields significant performance gains.

## 1 Introduction

Reinforcement Learning with verifiable rewards (RLVR) has been viewed as a promising way towards advanced reasoning LLMs, given the success of DeepSeek-R1(DeepSeek-AI et al., 2025) in mathematical reasoning and coding tasks. While RLVR is expected to benefit from the exploration-exploitation mechanism of RL to gain performance improvement, recent work(Nath et al., 2025; Yue et al., 2025) shows experimental evidence that popular RLVR algorithms, such as PPO (Schulman et al., 2017b), GRPO (Zheng et al., 2025), and even the more recent DAPO (Yu et al., 2025) that aims to encourage exploration, tend to gain performance improvement mainly from self-distillation, rather than from the 'capability gain' measured by the pass@k performance of high k value.

There have been many efforts to address this problem. Among them, one line of work tries to inject capability into the model by introducing extra offline data into the training process(Dong et al., 2025; Liu et al., 2025b). Specifically, given a training prompt, an extra response is provided by an explicit offline policy or from an offline dataset (which can be seen as sampled from an underlying behavior policy). Such practice can be seen as an extension of the typical SFT-RL pipeline of the LLM post-training. However, a fundamental drawback of these methods is that the offline behavior policy can no longer provide much help to the training when the online one surpasses its ability.

In this work, we show that for formal theorem proving – a task where the model response is highly structured and the extra info can be obtained during verification – we can efficiently boost the sample efficiency of RLVR training by exploiting the structural information in the

model proof. Specifically, we use the ProofAug method (Liu et al., 2025a) to repair part of the traces that fail to prove the theorems. It can be viewed as injecting the environment interaction information into the online policy to get an *associate policy*.

We first conduct a preliminary experiment showing that directly replacing the original proof with the ProofAug proof in RLVR can hurt performance. Then, we analyze the possible causes and conclude two principles, progress guarantee and variance reduction, to guide the algorithm design for applying ProofAug. Our derivations according to the principles lead to a novel policy optimization method **P**roximal **L**anguage modeling **P**olicy **O**ptimization (PLPO) that is applicable for training any autoregressive language model. Next, as for integrating the ProofAug into PLPO, we carefully select the variables to modify in the gradient update expression and add several conditions for applying the replacement of ProofAug, in order to avoid large extents of principle breaking.

We call the whole training pipeline described above by *ProofAug+*, and verify the effectiveness of it on the Goedel-Pset dataset. Our experimental results show that PLPO outperforms the baseline algorithms in stability and performance, and ProofAug+ further boosts the training efficiency. We conclude this section by emphasizing our contributions:

- We propose a novel RLVR training scheme for formal theorem proving, ProofAug+, which boosts the training process through integrating ProofAug into the RLVR pipeline in a conditioned way. ProofAug+ shows a margin of ~4.0% accuracy when compared with the GRPO-hybrid baseline.

- A novel RLVR training algorithm for LLM, Proximal Language Modeling Policy Optimization (PLPO), is proposed as the base algorithm of ProofAug+. Benefiting from using the exact policy advantage as the optimization target and employing a gradient rejection mechanism to substitute ratio-clipping, PLPO is superior in stability and performance than popular PPO variant counterparts.

- The ProofAug+ scheme serves as a theorem-proving domain example of boosting RLVR training sample efficiency with inference-time techniques. It is hopeful that it can inspire integration of techniques that share similar properties with ProofAug in other domains into the RLVR training.

## 2 Preliminaries

### 2.1 RLVR algorithms for LLMs

This subsection introduces popular Reinforcement Learning algorithms (with verifiable rewards) for LLMs and introduces our notations.

**Token-level RL.** Following Ouyang et al. (2022); Shao et al. (2024); DeepSeek-AI et al. (2025), the current mainstream practice of using RL for LLMs employs a token-level setting, where each token corresponds to an RL step. Tasks can be modeled as $T$-horizon MDPs when the number of maximum new tokens allowed to generate for the model is $T$. The state space is defined to be $\mathbb{V}^*$, the set of sequences with elements in $\mathbb{V}$, including the null string. Let $S_t$, $R_t$ denote the state and reward at time $t$, respectively. The specialness of the language modeling setting for RL lies in that the $t + 1$ time state $S_{t+1}$ is exactly the concatenation of the time-$t$ state and action. Thus, a trace $\tau$ is determined only by the states $S_0, S_1, \cdots, S_T$. For integrity, we assume that the LMs always output padding tokens till time $T$ when it stopped earlier than $T$. The environment reward is typically a sequence-level 0-1 reward dependent on a verifier, such as the Lean 4 proof checker in our case. For a token sequence $s \in \mathbb{V}^*$, the verifier outputs a binary value $R(s)$ indicating whether $s$ completes the task. In the token-level setting, the reward could only be non-zero at $t = T - 1$ and equals to $R(S_T)$.

When the initial state $S_0$ is drawn from an underlying theorem statement distribution $D \in \mathcal{P}(\mathbb{V}^*)$, the goal of RL is to learn a parameterized policy $\pi_\theta$ (the LLM, with fixed sampling strategy) to maximize the expected total return

$$\eta_D(\pi) := \mathbb{E}_{x \sim D}[\eta_x(\pi)], \tag{1}$$

where $\eta_x(\pi) := V_\pi(x) := \mathbb{E}_{\tau \sim \pi}[\sum_{t=0}^{T-1} R_t | S_0 = x]$ is the prompt-wise target (the value for $\pi$ at prompt $x$). To optimize Equation (1), policy gradient methods are the popular choices. Vanilla policy gradient, or REINFORCE (Sutton et al., 1999), performs gradient ascent on the target using the Monte-Carlo estimated gradient. Since the vanilla policy gradient may suffer from large variance and instability, subsequent methods propose to constrain the policy update in each iteration to address this issue. For example, the Proximal Policy Optimization (PPO) (Schulman et al., 2017b) method applies ratio-clipping on a surrogate objective and proposes to optimize the following target

$$J_\pi^{\mathrm{PPO}}(\tilde{\pi}) = \mathbb{E}_{s \sim \rho_\pi, a \sim \pi(\cdot|s)} \left[\min\left(r A_\pi(s, a), \mathrm{clip}(r, 1 - \epsilon_l, 1 + \epsilon_h) A_\pi(s, a)\right)\right], \qquad (2)$$

for each iteration. Here, $\rho_\pi(s) := \sum_{t=0}^{\infty} \gamma^t P_\pi(S_t = s)$ denotes the visiting frequency of $s$ when using the policy $\pi$ to sample the traces, $A_\pi(s, a) := V_\pi(s \oplus a) - V_\pi(s)$ is the advantage function, and $r := \frac{\tilde{\pi}(a|s)}{\pi(a|s)}$ is the importance ratio. estimated by generalized advantage estimation (GAE) (Schulman et al., 2018) method, using an extra parameterized value model. Recently, Shao et al. (2024) proposes GRPO, where the advantage is estimated by the group normalized reward $(R(S^i) - \mathrm{mean}(\{S^i\}_i^n))/\mathrm{Std}\{S^i\}_i^n$ for a group of samples $\{S^i\}_i^n$ from the same prompt to obviate the need of an extra value model.

**Sequence-level RL.** The recent work Zheng et al. (2025) argues for aligning the unit of the the reward and that of the optimization target, thus proposing the following sequence-level GSPO target

$$J_\pi^{\mathrm{GSPO}}(\tilde{\pi}) = \mathbb{E}_{y \sim \pi(\cdot|x)} \left[\min\left(r A_\pi(y), \mathrm{clip}(r, 1 - \epsilon_l, 1 + \epsilon_h) A_\pi(y)\right)\right], \qquad (3)$$

where $r := \left(\frac{\tilde{\pi}(y|x)}{\pi(y|x)}\right)^{\frac{1}{|y|}}$ and $y \sim \pi(\cdot|x)$ represents that $y$ is a response sampled from $\pi$ given the prompt $x$. The estimation of the advantage $A_\pi$ maintains the same with GRPO.

## 2.2 PRELIMINARY EXPERIMENT: NAIVE DIRECT-REPLACEMENT

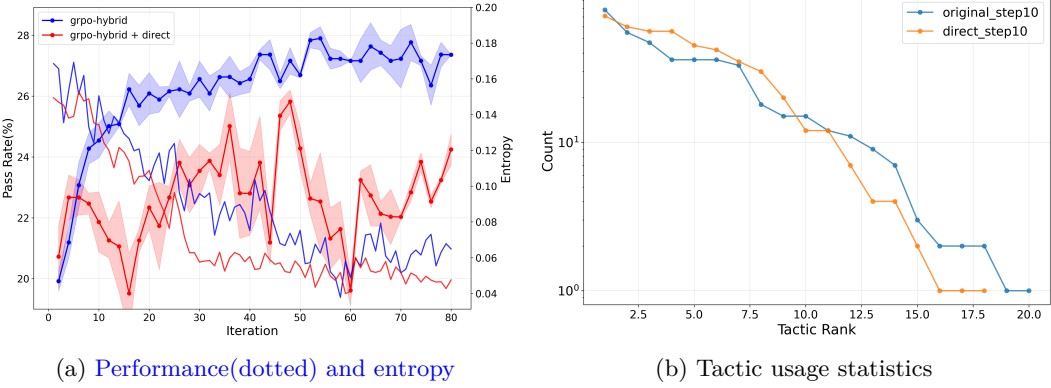

(a) Performance(dotted) and entropy  (b) Tactic usage statistics

Figure 1: Results of naive trace replacement using ProofAug

We first briefly introduce Lean and ProofAug here. Lean is an interactive theorem prover that supports verifying any proof to a statement by inferring whether it can be reduced to axioms. Given a proof attempt $y$ by LLM, if the attempt is not passing the Lean proof check, ProofAug is an operator $O$ that repairs it by trying a list of tactics on the intermediate conjectures in which the errors locate in a back-tracing way to provide a possibly repaired proof $y^O$(if no proof found, we assume $y^O = y$). See Section A for more details.

To integrate ProofAug into the RL pipeline, a naive idea is to replace all the wrong proof traces with corrected ones, by which we call 'direct-replacement'. We follow Lin et al. (2025b) to similarly use a hybrid GRPO-based algorithm that incorporates the Dr.GRPO advantage estimation method and removal of length normalization, and the zero-KL coefficient practice of DAPO into the GRPO pipeline. Formally, the empirical step gradient of GRPO-hybrid algorithm is

---

**Algorithm 1 PLPO**: **P**roximal **L**anguage Modeling **P**olicy **O**ptimization

---

**Require:** Max iteration $M$, Task prompt distribution $D$, rollout sample size $N$, sample batch size $B$, initial LLM $\pi_{\text{init}}$ and an optimizer (AdamW)
1: $\tilde{\pi} \leftarrow \pi_{\text{init}}$
2: **for** Iteration $k = 1, 2, \cdots$ **to** $M$ **do**
3:     $\pi \leftarrow \tilde{\pi}$
4:     **for** Batch $b = 1, 2, \cdots$ **to** $\lfloor \frac{N}{B} \rfloor$ **do**
5:        Sample a batch of prompts $D_b$ of size $B$
6:        For each prompt $x$, sample $n$ responses $\mathbf{y}_x = \{y^i\}_{i=1}^n$ according to $\pi$
7:        Get the rewards $\{R(y^i)\}_{i=1}^n$ from the verifier
8:        Calculate $g_b \leftarrow \frac{1}{Bn} \sum_{x \in D_b} \sum_{i \in [n]} g^i$ according to Equation (5)
9:        Use the optimizer to update $\tilde{\pi}$ given the current step gradient $g_b$
10:    **end for**
11: **end for**
12: Output $\tilde{\pi}$

---

$$\frac{1}{G} \sum_{i=1}^{G} \sum_{t=1}^{|y^i|} \left\{ \min \left[ \frac{\tilde{\pi}(y_t^i \mid x, y_{<t}^i)}{\pi(y_t^i \mid x, y_{<t}^i)} \hat{A}_t^i, \ \text{clip}\left( \frac{\tilde{\pi}(y_t^i \mid x, y_{<t}^i)}{\pi(y_t^i \mid x, y_{<t}^i)}, 1 - \epsilon, 1 + \epsilon \right) \hat{A}_t^i \right] \right\} \quad (4)$$

for the group of responses $\{y_i\}_{i=1}^n$ sampled from the problem prompt $x$, with the Leave-one-out(LOO) advantage estimation $\hat{A}_t^i = R(y^i) - \frac{\sum_{j \neq i} R(y^j)}{n-1}$. For direct-replacement, all the traces and their rewards are replaced with their ProofAug counterparts and the corresponding rewards.

We show the direct-replacement experimental results in Figure 1. The lines with dots are test performance, and those without dots are entropy curves. It can be seen that adding ProofAug naively not only is detrimental to both the stability and performance, but also accelerates the entropy collapse. We also collect statistics on the tactics the models use after 10 steps training in these two settings in Figure 1b. It can be seen that after direct-replacement, the model tends to use fewer tactics. We call this phenomenon 'tactic collapse'.

## 3 METHOD

To handle the issues observed in the preliminary experiment, the design of ProofAug+ is guided by the following two principles: 1)**Progress Guarantee.** There should be a theoretical guarantee or at least a convincing informal argument that the objective Equation (1) is expected to increase in each policy optimization iteration. In other words, the optimization target should better be an unbiased estimation of the expected total return of the target policy, which requires practice that breaks the estimation such as direct-replacement to be suppressed. Besides, entropy collapse should be mitigated to encourage exploration, in order that the training can still make progress in the long-term, which requires that teaching the model to use several fixed tactics like in the direct-replacement experiment should be avoided. 2)**Variance Reduction.** The variance of the gradient estimation should be reduced to increase the stability and accelerate the training. Factors such as outlier samples or abrupt gradient updates leading to the overfitting should be suppressed. **blue**For example, in the direct-replacement, the importance ratio is no longer the ratio of target policy and the real sampling policy, which could make the absolute value of the gradient step too large.

In this section, we will first review the derivation of PPO and propose to make several modifications according to the principles, obtaining our PLPO algorithm tailored for LLM training, as shown in Algorithm 1[1]. The PLPO sample gradient is:

$$g^i = \left( 1 - \mathbb{1}_{\widehat{\text{GRA}}}(y^i; \pi, \tilde{\pi}) \right) \frac{\tilde{\pi}(y^i|x)}{\pi(y^i|x)} \hat{A}_\pi(y^i) \nabla \log \tilde{\pi}(y^i|x), \quad (5)$$

---

[1]For simplicity, we only describe the algorithm for the case of only 1 epoch in each iteration.

---

**Algorithm 2 ProofAug+**

---

**Require:** Max iteration $M$, Task prompt distribution $D$, rollout sample size $N$, sample batch size $B$, initial LLM $\pi_{\text{init}}$, an optimizer (AdamW), a ProofAug operator $O$ and a ProofAug application criteria $C$ (Equation (9))

1: $\tilde{\pi} \leftarrow \pi_{\text{init}}$
2: **for** Iteration $k = 1, 2, \cdots$ **to** $M$ **do**
3:    $\pi \leftarrow \tilde{\pi}$
4:    **for** Batch $b = 1, 2, \cdots$ **to** $\lfloor \frac{N}{B} \rfloor$ **do**
5:       Sample a batch of prompts $D_b$ of size $B$
6:       For each prompt $x$, sample $n$ responses $\{y^i\}_{i=1}^n$ according to $\pi$
7:       Apply ProofAug on the samples to get $\{y^{iO}\}_{i=1}^n \leftarrow \{O(y^i)\}_{i=1}^n$
8:       Get the rewards $\{R(y^i)\}_{i=1}^n$ and $\{R(y^{iO})\}_{i=1}^n$ from the verifier
9:       Calculate $C^i \leftarrow C(y^i; y^{iO}, \{R(y^j)\}_{j=1}^n)$ according to Equation (9)
10:      Calculate $g_b \leftarrow \frac{1}{Bn} \sum_{x \in D_b} \sum_{i \in [n]} g^{iO_C}$ according to Equation (8)
11:      Use the optimizer to update $\tilde{\pi}$ given the current step gradient $g_b$
12:    **end for**
13: **end for**
14: Output $\tilde{\pi}$

---

where we use the LOO advantage estimation $\hat{A}_\pi(y^i) = R(y^i) - \frac{\sum_{j \neq i} R(y^j)}{n-1}$, and GRA stands for the *gradient rejection area*,

$$\widehat{\text{GRA}}(y^i; \pi, \tilde{\pi}) = (\hat{A}_\pi(y^i) > 0 \wedge r(y^i) > 1 + \epsilon_h) \vee (\hat{A}_\pi(y^i) < 0 \wedge r(y^i) < 1 - \epsilon_l), \quad (6)$$

where we provide two types of rejection indicator $r$, 'sum' and 'average', for more flexible control of rejection area than ratio-clipping:

$$r(y) := \begin{cases} \frac{\tilde{\pi}(y|x)}{\pi(y|x)}, & \text{sum-type,} \\ \left(\frac{\tilde{\pi}(y|x)}{\pi(y|x)}\right)^{\frac{1}{|y|}}, & \text{average-type.} \end{cases} \quad (7)$$

Based on PLPO, given a ProofAug operator $O$ on $\mathbb{V}^T$ that transforms $y^i$ to $y^{iO}$, we set a list of conditions for applying $O$ on $y^i$ in ProofAug+, represented by a boolean value function $C(y^i; y^{iO}, \{R(y^j)\}_{j=1}^n)$, abbreviated by $C^i$ for the $i$th sample. The sample gradient in ProofAug+ is:

$$g^{iO_C} := \left(1 - 1_{\widehat{\text{GRA}}}(y^{iO_C}; \pi, \tilde{\pi})\right) \frac{\tilde{\pi}(y^{iO_C}|x)}{\pi(y^{iO_C}|x)} \nabla \log \tilde{\pi}(y^{iO_C}|x) \hat{A}_\pi(y^{iO_C}), \quad (8)$$

where $\hat{A}_\pi(y^{iO_C}) = R(y^{iO_C}) - \frac{\sum_{j \neq i} R(y^j)}{n-1}$ as before, and $y^{iO_C} = C^i y^{iO} + (1 - C^i) y^i$. The specific form of $C$ is:

$$C(y^i; y^{iO}, \{R(y^j)\}_j^n) = \begin{cases} \text{True} & \text{if } d(y^{iO}) \neq 1 \text{ and } d(y^{iO}) \geq d(y^i) \text{ and } \forall j, R(y^j) = 0, \\ \text{False} & \text{otherwise,} \end{cases} \quad (9)$$

where $d(y)$ is the depth of the proof.

The rest of the section states how PLPO and ProofAug+ are derived and explains the design choices.

## 3.1 Derivation of PLPO

As stated above, PLPO is derived from PPO with modifications tailored for LLM under our two principles. Let us review the derivation of PPO in the token-level setting first.

### Reviewing the derivation of TRPO/PPO

The optimization target for TRPO and PPO (before the ratio-clipping operation) is derived in an RL setting with a discounting factor $\gamma < 1$. Given a stochastic policy $\pi$, the following

identity (proved in Kakade & Langford (2002)) helps express the expected return $\eta(\tilde{\pi})$ of another policy $\tilde{\pi}$ in terms of $\pi$: $\eta(\tilde{\pi}) = \eta(\pi) + \sum_s \rho_{\tilde{\pi}}(s) \sum_a \tilde{\pi}(a|s) A_\pi(s, a)$.

Schulman et al. (2017a) shows a lower bound for the *policy advantage* $\eta(\tilde{\pi}) - \eta(\pi)$:

$$\eta(\tilde{\pi}) - \eta(\pi) \geq J_\pi^{\mathrm{CPI}}(\tilde{\pi}) - \frac{4\gamma \max_{s,a} |A_\pi(s, a)|}{(1 - \gamma)^2} D_{\mathrm{KL}}^{\max}(\pi, \tilde{\pi}),$$

where the *conservative policy iteration* (Kakade & Langford, 2002) objective $J_\pi^{\mathrm{CPI}}(\tilde{\pi}) = \sum_s \rho_\pi(s) \sum_a \tilde{\pi}(a|s) A_\pi(s, a)$ is a quantity that does not include a $\rho_{\tilde{\pi}}(\cdot)$ term and is thus used as the surrogate objective in TRPO and PPO.

In this work, we point out that for the LM question-response task, where we have a fixed horizon $T$ and $\gamma = 1$, and $\rho_\pi(s) = \pi(s) := \prod_{t=0}^{T-1} \pi(s_{t+1}|s_{:t})$, there is no need to use the surrogate objective $J_\pi^{\mathrm{CPI}}(\tilde{\pi})$. Instead, we have the following lemma:

**Lemma 1.** *Under the token-level LLM RL setting described in Section 2.1, the policy advantage $\eta(\tilde{\pi}) - \eta(\pi) =: J_\pi^{LM}(\tilde{\pi})$ has the following form:*

$$J_\pi^{LM}(\tilde{\pi}) := \eta(\tilde{\pi}) - \eta(\pi) = \mathbb{E}_{s \sim \rho_q, a \sim q} \left[ \frac{\tilde{\pi}(s \oplus a|x)}{q(s \oplus a|x)} A_\pi(s, a) \right], \tag{10}$$

*and the corresponding gradient is*

$$\nabla J_\pi^{LM}(\tilde{\pi}) = \mathbb{E}_{s \sim \rho_q, a \sim q} \left[ \frac{\tilde{\pi}(s \oplus a|x)}{q(s \oplus a|x)} \nabla \log \tilde{\pi}(s \oplus a|x) A_\pi(s, a) \right], \tag{11}$$

*where $q(\cdot|\cdot) : \mathbb{V} \times \mathbb{V}^* \to \mathbb{R}^+$ is any sampling LM whose support contains the support of $\tilde{\pi}$. Then, under an infinitesimal learning rate, the expected total return is expected to increase for a gradient step when the number of training samples approaches infinite and makes a good estimation of this expected gradient.*

We leave the proof of the lemma to Section C. The lemma can immediately lead to a policy optimization algorithm as long as we take the old policy $\pi$ as $q$ and determine an advantage estimation method. Nevertheless, the resulted algorithm could be instable due to the large updates. Following the variance reduction principle, we seek for a technique for suppressing large updates.

GRADIENT REJECTION MECHANISM

Although clipping the importance ratio is the standard PPO practice, we propose to decouple the target gradient estimation and large update suppression via a *gradient rejection* mechanism. We show our arguments below.

First note that the ratio-clipping operation equals to setting the sample gradients falling in the area

$$\{(s, a)|((A_\pi(s, a) > 0) \wedge (r(s, a) > \epsilon_{\mathrm{high}})) \vee ((A_\pi(s, a) < 0) \wedge (r(s, a) < \epsilon_{\mathrm{low}}))\}, \tag{12}$$

to zero, in order to control the policy update in one iteration. In PPO and its variants such as GRPO, $r$ is the importance ratio $\frac{\tilde{\pi}(a|s)}{\pi(a|s)}$. As to the case of Equation (11), if we follow-suit to use $\frac{\tilde{\pi}(s \oplus a)}{q(s \oplus a)}$ as the gradient rejection indicator, an obvious issue is that for long sequences the ratio might accumulate as the number of token increases and could make the rejection rate higher than expected.[2]

In this work, we argue that there is no a priori reason to choose a term that serves for the estimation of the policy advantage as the criteria for large updates rejection. Instead, we provide two choices for the update rejection criteria, 'average' and 'sum',

$$r(s, a) := \begin{cases} \frac{\tilde{\pi}(s \oplus a|x)}{\pi(s \oplus a|x)}, & \text{sum}, \\ \left(\frac{\tilde{\pi}(s \oplus a|x)}{\pi(s \oplus a|x)}\right)^{\frac{1}{|s| - |x| + 1}}, & \text{average}, \end{cases} \tag{13}$$

---

[2]A simple ideal experiment is to assume all token-wise ratio subjects to the Gaussian distribution, thus the random walk makes the sequence ratio explodes.

to meet the need of different training scenarios. The 'average' choice corresponds to the length normalized importance ratio used in Zheng et al. (2025), and the 'sum' choice degrades the target into a ratio-clipping loss.

Thus, the final form of the (expected) token-level PLPO gradient is

$$\nabla J_\pi^{\text{PLPO-token}}(\tilde{\pi}) = \mathbb{E}_{s \sim \rho_q, a \sim q} \left[ (1 - \mathbb{1}_{\text{GRA}^{\text{token}}}(s, a; \pi, \tilde{\pi})) \frac{\tilde{\pi}(s \oplus a)}{q(s \oplus a)} \nabla \log \tilde{\pi}(s \oplus a) A_\pi(s, a) \right],$$
(14)

where the token-level GRA is defined by

$$\text{GRA}^{\text{token}}(s, a; \pi, \tilde{\pi}) := (A_\pi > 0 \wedge r(s, a) > 1 + \epsilon_h) \vee (A_\pi < 0 \wedge r(s, a) < 1 - \epsilon_l).$$

### Transfer to sequence-level setting

In Equation (14), the token-level advantage $A_{\pi,t}^i(y_{:t}^i, y_{t+1}^i)$ is hard to estimate without a value model. The group-relative advantage estimation in GRPO and subsequent work such as Dr. GRPO (Liu et al., 2025c) directly use the sequence-level advantage estimations, for example, the LOO estimation $R_{\text{LOO}}^i := R(S^i) - \frac{1}{n-1} \sum_{j \neq i} R(S^j)$, to substitute the token-level advantage $A_{\pi,t}^i(y_{:t}^i, y_{t+1}^i) = V_\pi(y_{:t+1}^i) - V_\pi(y_{:t}^i)$ for all $t$ values. Although $R(S^i)$ is an estimation of $V_\pi(y_{:t+1}^i)$ when conditioned on the filtration before $t + 1$, $\frac{1}{n-1} \sum_{j \neq i} R(S^j)$ is not correlated to $V_\pi(y_{:t}^i)$ unless the other traces share identical prefix with $y^i$.

As a result, we decide to transfer to the sequence-level setting to avoid this obstacle. To transfer Equation (14) to the sequence-level setting, it suffices to use the whole response $y$ to substitute the token action $a$ and the only state $s$ that will be sampled is the initial state $S_0 = x$. We obtain

$$\nabla J_\pi^{\text{PLPO}}(\tilde{\pi}) = \mathbb{E}_{y \sim \pi(\cdot|x)} \left[ (1 - \mathbb{1}_{\text{GRA}}(y; \pi, \tilde{\pi})) \frac{\tilde{\pi}(y)}{\pi(y)} \nabla \log \tilde{\pi}(y) A_\pi(y) \right],$$
(15)

with the GRA

$$\text{GRA}(y; \pi, \tilde{\pi}) := (A_\pi(y) > 0 \wedge r(y) > 1 + \epsilon_h) \vee (A_\pi < 0 \wedge r(y) < 1 - \epsilon_l)$$
(16)

and $r(y)$ defined in Equation (7). We note that under the sequence-level RL setting, the LOO estimation is unbiased for the sequence-level advantage $A_\pi(y) = V_\pi(x \oplus y) - V_\pi(x)$. The practical algorithm corresponding to Equation (15) is exactly the PLPO algorithm shown in Algorithm 1.

### 3.2 Integrating ProofAug into PLPO

Starting from PLPO, we would like to incorporate PLPO in a way that also preserves the two principles. The naive direct-replacement approach under PLPO can be interpreted as using $\pi_O$ as the rollout distribution for estimating the policy advantage, i.e. trying to calculate

$$\left(1 - \mathbb{1}_{\widehat{\text{GRA}}}(y^{iO}; \pi_O, \tilde{\pi})\right) \frac{\tilde{\pi}(y^{iO}|x)}{\pi_O(y^{iO}|x)} \hat{A}_{\pi_O}(y^{iO}) \nabla \log \tilde{\pi}(y^{iO}|x),$$
(17)

while directly using $\pi(y^{iO})$ to estimate $\pi_O(y^{iO})$, thus causing a distribution shift in the estimation of the importance ratio, breaking the progress guarantee principle. At the same time, it also results in inappropriate gradient rejection rule, which could lead to large update breaking the variance reduction rule. As a result, despite that $\pi_O$ is expected to explore more positive reward traces, the training performance degrades.

**Conditioned proof repair.** To mitigate the distribution shift when estimating the PLPO $\pi_O$-sample gradient, we choose to add constraints for applying ProofAug to strike a balance between additional positive reward signals and distribution shift, i.e. the repaired proofs that we infer do not benefit the training too much are abandoned. Specifically, we set the following restrictions for applying the ProofAug operation $O$ on a proof response $y^i$: 1)$d(y^{iO}) \neq 1$ and $d(y^{iO}) \geq d(y^i)$. We reject the ProofAug proofs with depth=1, i.e. the single tactic case, and cases where ProofAug decreases the depth of the original proof.

2)$\forall j \in [n], R(y^j) = 0$. This conservative strategy requires we only apply ProofAug when all the original proofs fail. The boolean value function $C(y^i; y^{iO}, \{R(y^j)\}_{j=1}^n)$ indicates whether the above conditions are satisfied. The explicit expression of $C$ is given in Equation (9).

We explain the design of two rules as follows:(1)Depth-decrease proofs discourage the model to think deep. It only tells the model to use the high-level automated tactic (2)When we can already sample an $R(y) = 1$ trace from the prompt, learning from this proof is expected to inject most of the key knowledge related to the statement to the model. As a result of the two constraints, we can reduce the frequency of ProofAug application, thus suppressing the distribution shift and tactic collapse without satisfying the useful positive reward signal that can teach the model novel proving patterns.

There is another difference between our final ProofAug+ algorithm and the one corresponding to Equation (17). Equation (17) is derived by using $\pi_O$ as the Monte-Carlo estimation sampling distribution. As a result, the LOO estimation of the value baseline uses $\{R(y^{jO})\}_{j \neq i}$ instead of the original reward baseline $\frac{\sum_{j \neq i} R(y^j)}{n-1}$. In order to let $\tilde{\pi}$ learn not only from the actions that increase $V_{\pi_O}$, but also from what $\pi_O$ can prove while $\pi$ cannot, we choose to use the original reward baseline.

Integrating the techniques introduced in this section, including PLPO and conditioned proof repair, we finally obtain our ProofAug+ pipeline, as described in Algorithm 2.

## 4 EXPERIMENT

### 4.1 EXPERIMENTAL SETUP

**Dataset and Base model** In this work, we validate our claims by training Qwen2.5-1.5B-Instruct (Qwen et al., 2025) on Goedel-Pset (Lin et al., 2025a), the largest open-source Lean 4 statement dataset covering from high school exercises to Olympiad-level problems, and test the trained model on a subset of 497 samples of Goedel-Pset. We first sample 10k problems from Goedel-Pset and use Kimina-Prover-1.7B (Wang et al., 2025) to prove them. Among the 10k samples, 8844 statement-proof pairs (with the thinking process removed) can be extracted. We use the 8844 data pairs as the cold-start data to fine-tune Qwen2.5-1.5B-Instruct. The obtained model is the initial model for our Reinforcement Learning.

**Base training setting.** We perform RL starting from the initial model using the rest of Goedel-Pset (of around 1.72M problems) as the prompt set with a constant learning rate 1e-6. We use OpenRLHF (Hu et al., 2024) as the training framework and use the async training mode with async queue size equal to 1. Our base strategy is **grpo-hybrid**, where we set the clip ratios as $\epsilon_l = 0.2$ and $\epsilon_h = 0.28$ and set the KL coefficient to 0 following DAPO (Yu et al., 2025), and use an LOO estimation for the advantage following Liu et al. (2025c). For **plpo**, we use the sum-type gradient rejection by default, with $\epsilon_l = 0.2$ and $\epsilon_h = 0.28$. For **gspo**, we follow Zheng et al. (2025) to use $\epsilon_l = 0.2, \epsilon_h = 0.27$, but still use the LOO estimation. In each policy iteration, we sample $n = 8$ samples with a temperature $T = 0.6$ for each of the $N = 64$ prompts in a rollout. For each rollout, we train for 1 epoch, and each training batch contains $B = 8$ prompts. The maximum number of tokens for model response is 3072. All trainings are done on 8 Nvidia RTX 3090 GPUs with 4 GPUs for sampling and the other 4 are used for 4 policy actors.

**Lean and ProofAug setting.** We build a lean server based on the Lean REPL v4.20.0 tailored for tactic-wise execution for ProofAug. The server also includes cache systems like in the Kimina server(Santos et al., 2025) to accelerate the verification process to meet the need of RLVR training efficiency. During training, there is a 120s timeout for each step in Lean and a 180s limit for the whole verification process including the first-pass compilation and the ProofAug stage. Also, for encouraging the tactic diversity, we shuffle the order of the heuristic tactics to use for ProofAug.

## 4.2 RESULTS

**PLPO enjoys stability and better long-run performance** Figure 2 shows both the pass@1 accuracy curve and the entropy curve for comparing PLPO and GRPO-hybrid in our setting.

**ProofAug+ achieves clear margin over the baseline algorithm.** Figure 2 also shows that the improvement of ProofAug+ comes from both the PLPO algorithm and the conditioned proof repair introduced in Section 3.2. The performance gain compared to the peak accuracy of GRPO-hybrid is around 4.0%.

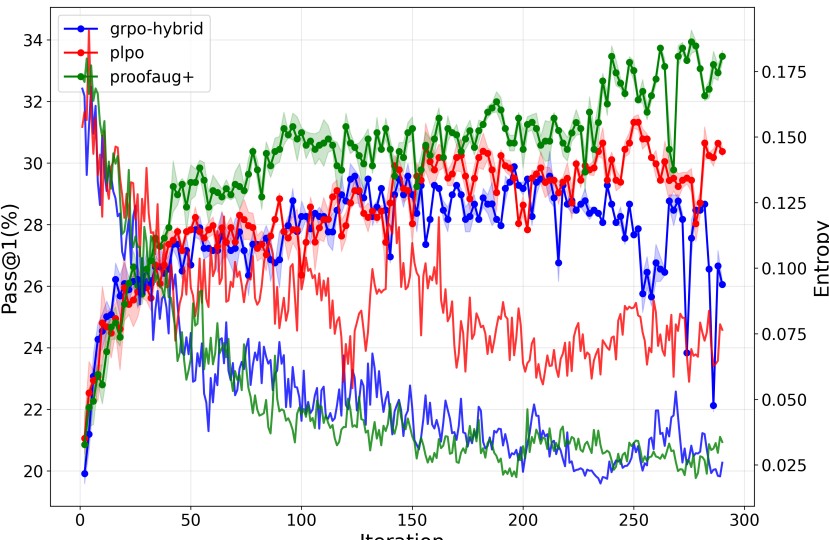

Figure 2: The performance of ProofAug+ compared with baseline algorithm GRPO-hybrid

**Decoupling gradient estimation and update rejection helps with stability.** In Figure 3, the original GSPO algorithm leads to instability and entropy collapse during training in our task. This is because GSPO is tailored for long-sequence thinking models, while our task focuses on directly generating proofs of code. In contrast, when using the sum-type gradient rejection rule described in Equation (6), the GSPO training then goes well.

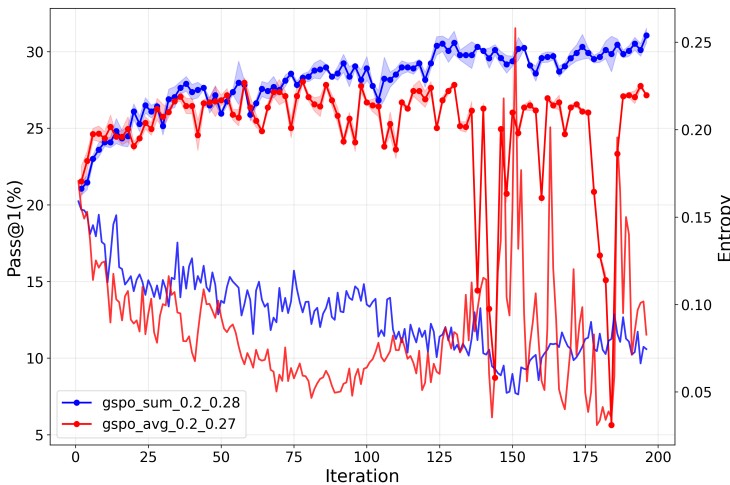

Figure 3: The original GSPO vs. GSPO with sum-type gradient rejection

For extensive experimental results, see Section D.

## 5 CONCLUSION AND LIMITATIONS

In conclusion, this work proposes practical solutions for the failure of naive direct-replacement integration of ProofAug into RLVR, guided by the progress guarantee and variance reduction principles. The proposed PLPO algorithm is a general sequence-level algorithm for LLM RL training, and the ProofAug+ pipeline is inspiring for designing schemes for integrating inference-time techniques in other domains.

Nevertheless, there are some limitations for this work. First, this work has focused the case where the sampling strategy of the LLM during training completely matches that in the evaluation time, to rule out the factors led by train-test sampling mismatch. As to popular pass@k evaluations, we remain the integration of ProofAug with pass@k training (Walder & Karkhanis, 2025; Chen et al., 2025) for future work. Second, it is important to try PLPO on different experimental setups, since the optimization behaviors on thinking models and other domains might be very different. Finally, this work only includes derivations of the algorithms according to the guiding principles, lacking rigorous convergence analysis for them, which we also remain for future work.

## Reproducibility statement

The experiments in this paper are reproducible by following the setup in Section 4.1 and the reproducibility-related details stated in Section E. The proofs of the lemmas in the main text can be found in Section C.

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

## A  An introduction to Lean and ProofAug

Lean is one popular proof assistant grounded in dependent type theory, which provides more powerful features than simply typed systems. Also, Lean (from version 4) itself is a functional programming language that can serve for general programming tasks and allows metaprogramming just in the same environment of common usage (which is called *reflection*) for users to flexibly build their own syntax and tactics. Moreover, the theory library for Lean, Mathlib (mathlib Community, 2020), containing theorems from a large range of fields including Algebra, Combinatorics, Analysis and Probability and is continuously growing.

Nevertheless, there is currently no built-in hammer tools such as Sledgehammer for Lean yet, making the proof writing in Lean troublesome in some cases. Fortunately, some tactics provide by Lean, such as `simp, bound, omega, linarith` and the in-development `grind` are also very strong and are keep growing, making Lean an easier-to-use language as the time goes.

An example of Lean 4 proof generated by Kimina-Prover (Wang et al., 2025), a state-of-the-art level LLM theorem prover, is shown below:

```
theorem amc12a_2021_p3 (x y:ℕ) (h0:x + y = 17402) (h1: 10|x) (h2: x/10=y):
    ↑x-↑y=(14238: ℤ) := by
  have h3: x = 10 * y := by
    have h6: x / 10 = y := h2
    have h7: x = 10 * (x / 10) := by
      omega
    rw [h6] at h7
    omega
  have eq1: 11 * y = 17402 := by
    omega
  have h8: y = 1582 := by
    omega
  have h9: x = 15820 := by
    omega
  omega
```

It can be seen that the proof is in a hybrid procedural and declarative style, containing rich hierarchical proof structure that will be very useful in our methods. ProofAug designs a pre-parser that identify keywords such as **by** as the indicator of the start of a block and infer from the number of indents whether a line of Lean code ends the block. Viewing each block as a subtree, we can then identify the tree structure.

Given the tree structure of a proof, the ProofAug procedure (without their ERP module) is illustrated in Figure 4 (modified from Liu et al. (2025a)). Briefly speaking, ProofAug first derives all potential coarse-grained proof sketches, i.e. proofs with 'sorry' placeholders that leave the proof to be completed later, from the initial proof proposal generated by the LLM theorem prover, then use a hammer tool or heuristic tactics like in Jiang et al. (2023).

## B  Related Work

**RLVR for theorem provers.** The technique reports Lin et al. (2025b); Wang et al. (2025) use RLVR training pipeline to enhance the performance after data collection. However, they only provide the weights of models, rather than fully open-source the whole training pipeline and data. In contrast, Our ProofAug+ method is fully open-sourced and we also release a Lean server to contribute to the community.

**Sequence level Reinforcement Learning algorithms.** GSPO (Zheng et al., 2025) is the most similar algorithm with PLPO among the sequence-level optimization methods. However, it is not derived from the theory, thus there is no direct guarantee of step progress for the total return. Team et al. (2025) uses the online mirror descent algorithm for policy optimization, using the relative entropy as the regularization instead of the loss-clipping technique or our gradient rejection rule.

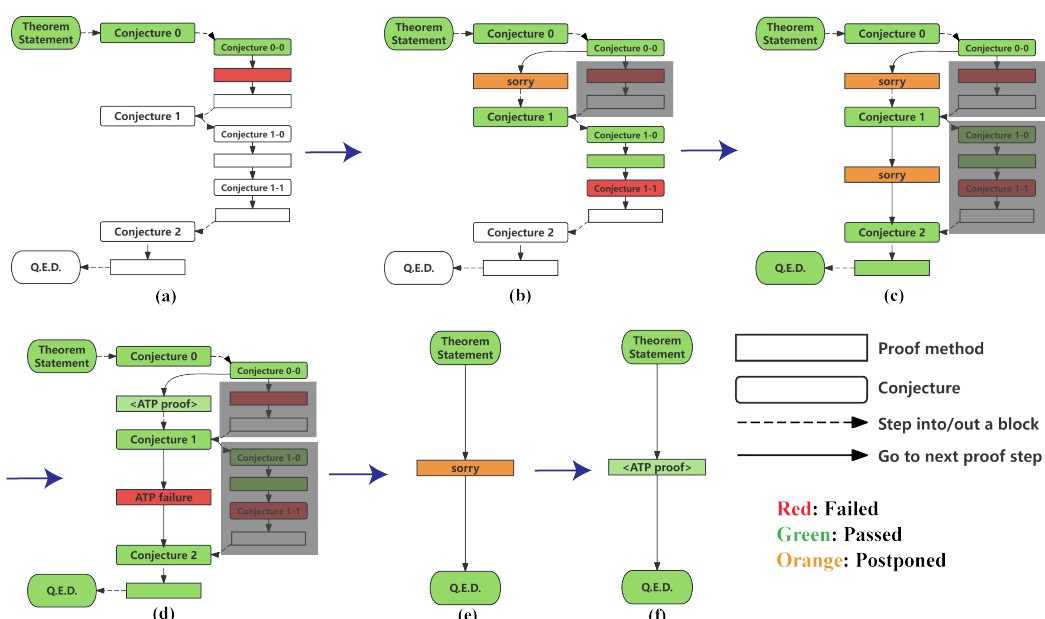

Figure 4: **An illustration flow of ProofAug.** Each box in the flowchart corresponds to a proof step. **(a)** The initial proof encounters an error when proving Conjecture 1-0. **(b)** We replace the original proof of Conjecture 1-0 with a **sorry** and continue, until a syntactic error occurs at Conjecture 1-1. **(c)** The proof of Conjecture 1 is replaced by **sorry**, and we obtain a semi-proof with two **sorry**. **(d)** Automation tools are called to prove Conjecture 0 and Conjecture 1, but failing at the latter one. **(e)** We resort to a more coarse level of proof. **(f)** Finally, this time we successfully find a proof with automation tools.

**Reinforcement Learning with offline policy.** Liu et al. (2025b) interpolates the cross-entropy loss and policy gradient surrogates, but no theoretical guarantee or analysis are made. Dong et al. (2025) also aims at countering the model capability in RLVR and provides an estimate of the underlying behavior policy for the offline data. In contrast to these works, ProofAug propose the concept of associate policy, which depends on the environment feedback, equipping algorithms like PLPO+ProofAug with the potential of steadily performance boost.

## C  PROOFS OF LEMMAS

To prove Lemma 1, we first need to prove the sequence-level version of policy advantage expression:

**Lemma 2.** *For the token-level RL setting introduced in Section 2, i.e. with horizon $T$, $\gamma = 1$ and $S_{t+1} = S_t \oplus A_t$, we have*

$$\eta(\tilde{\pi}) = \eta(\pi) + \sum_s \rho_{\tilde{\pi}}(s) \sum_a \tilde{\pi}(a|s) A_\pi(s, a), \tag{18}$$

*Proof.* On one hand, we have

$$\mathbb{E}_{\tau \sim \tilde{\pi}} \left[ \sum_{t=0}^{T-1} A_\pi(S_t, s_{t+1}) \right] = \mathbb{E}_{\tau \sim \tilde{\pi}} \left[ \sum_{t=0}^{T-1} R(s_t) + V_\pi(S_{t+1}) - V_\pi(S_t) \right]$$

$$= \mathbb{E}_{\tau \sim \tilde{\pi}} \left[ V_\pi(S_T) - V_\pi(S_0) + \sum_{t=0}^{T-1} R(S_t) \right] \tag{19}$$

$$= -\mathbb{E}_{s_0} \left[ V_\pi(s_0) \right] + \mathbb{E}_{\tau | \tilde{\pi}} \left[ \sum_{t=0}^{\infty} \gamma^t r(s_t) \right]$$

$$= -\eta(\pi) + \eta(\tilde{\pi}).$$

On the other hand, we have

$$\mathbb{E}_{\tau | \tilde{\pi}} \left[ \sum_{t=0}^{T-1} A_\pi(s_t, a_t) \right] = \sum_s \rho_{\tilde{\pi}}(s) \sum_a \tilde{\pi}(a|s) A_\pi(s, a), \tag{20}$$

$\square$

*Proof of lemma Lemma 1.* According to Lemma 2, we have

$$\eta(\tilde{\pi}) - \eta(\pi) = \sum_s \tilde{\pi}(s) \sum_a \tilde{\pi}(a|s) A_\pi(s, a)$$

$$= \sum_s q(s) \frac{\tilde{\pi}(s)}{q(s)} \sum_a q(a|s) \frac{\tilde{\pi}(a|s)}{q(a|s)} A_\pi(s, a)$$

$$= \sum_s \rho_q(s) \sum_a q(a|s) \frac{\tilde{\pi}(s \oplus a)}{q(s \oplus a)} A_\pi(s, a) \tag{21}$$

$$= \mathbb{E}_{s \sim \rho_q, a \sim q} \left[ \frac{\tilde{\pi}(s \oplus a)}{q(s \oplus a)} A_\pi(s, a) \right],$$

where all summations are taken on the support of $\tilde{\pi}$. As to the gradient, by exchanging the expectation and the $\nabla$ (where we have assumed $q$ satisfies regularity conditions that controls $\|\frac{\nabla \tilde{\pi}}{q}\|$) and using $\nabla \tilde{\pi} = \tilde{\pi} \nabla \log \tilde{\pi}$, we have

$$\nabla J_\pi^{\text{LM}}(\tilde{\pi}) = \mathbb{E}_{s \sim \rho_q, a \sim q} \left[ \frac{\tilde{\pi}(s \oplus a|x)}{q(s \oplus a|x)} \nabla \log \tilde{\pi}(s \oplus a|x) A_\pi(s, a) \right] \tag{22}$$

as desired. $\square$

## D   ADDITIONAL EXPERIMENTAL RESULTS

**Tactic Usage Statistics.** Figure 5 shows that our conservative strategy can efficiently encourage the usage of more tactics, avoiding policy collapse.

**Tactic distribution.** We show it here for comparison with Figure 6. It can be seen that the tactics used by ProofAug come to the top usages.

**Results on miniF2F.** We also test the performance of the trained models on the cross-language benchmark miniF2F (Zheng et al., 2021). It turns out that while PLPO works well, adding the conservative strategy ProofAug is not beneficial for generalization to out-of-distribution data. This result also calls for a more diverse and high-quality open-source formal statement dataset.

**More Comparison between PLPO and other algorithms.** We show extra experiments for PLPO comparison in Figure 8.

**Comparison of ProofAug+ and GRPO-hybrid with conditioned proof repair.** It is shown in Figure 9 that ProofAug+, which equals to PLPO with conditioned proof repair, has better stability than GRPO-hybrid with conditioned proof repair.

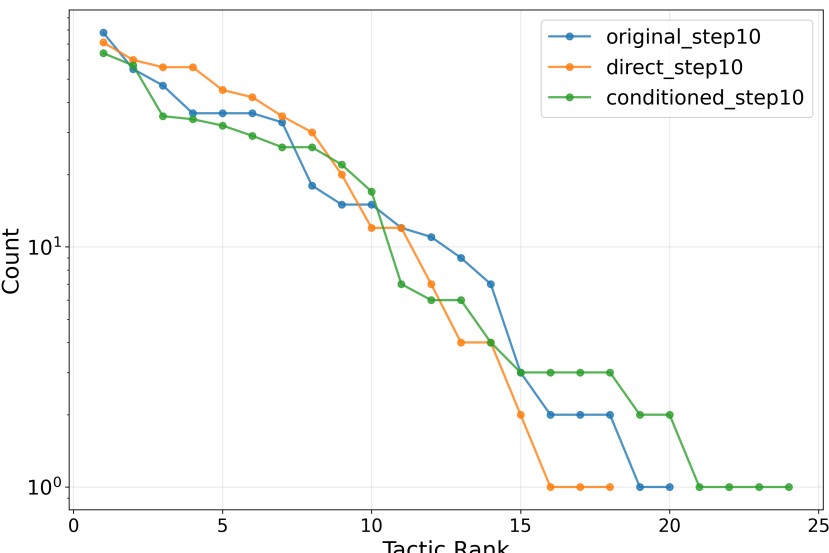

Figure 5: ProofAug+ encourages diverse tactic usage

**Different sizes of models.** Figure 10 shows the experiments on Qwen2.5-0.5B-Instruct with the same procedure described in Section 4.1. It turns out that the advantage of ProofAug+ over ProofAug is consistent with the results shown in Figure 2, around 2%.

## E  REPRODUCIBILITY DETAILS

**Data Processing of Goedel-pset.** We download the Goedel-pset from their huggingface repo[3]. Then we sample 10k samples for training the initial model for RL and 497 samples (that has no coverage with the 10k) for evaluation. The remained are used for training. For the training prompt construction, we follow the non-CoT template of DeepSeek-Prover-V2-7B (Ren et al., 2025) to construct messages and then use the Qwen2.5-Instruct built-in chat template to translate to a prompt.

**OpenRLHF setting.** We turn on the use of –full-determination in our experiments and always use the default seed when using the OpenRLHF train_ppo_ray script for our experiment.

**Evaluation and Lean setting.** We use evaluate any model/checkpoint three times with different seeds to provide an error bar at each data point. The result is only reproducible for the specific Lean and Lean REPL version. We also observe that since some tactics of Lean such as `omega` include CPU-heavy searching procedures which may make the verification result might differ for CPUs of different performance. Nevertheless, since we have set a 120s timeout for each Lean step, this situation should rarely happen.

## F  THE USAGE OF LARGE LANGUAGE MODELS

During writing this paper, we have used commercial LLMs in helping polishing the paragraphs after we first write a draft and detecting grammar errors.

---

[3]https://huggingface.co/datasets/Goedel-LM/Goedel-Pset-v1

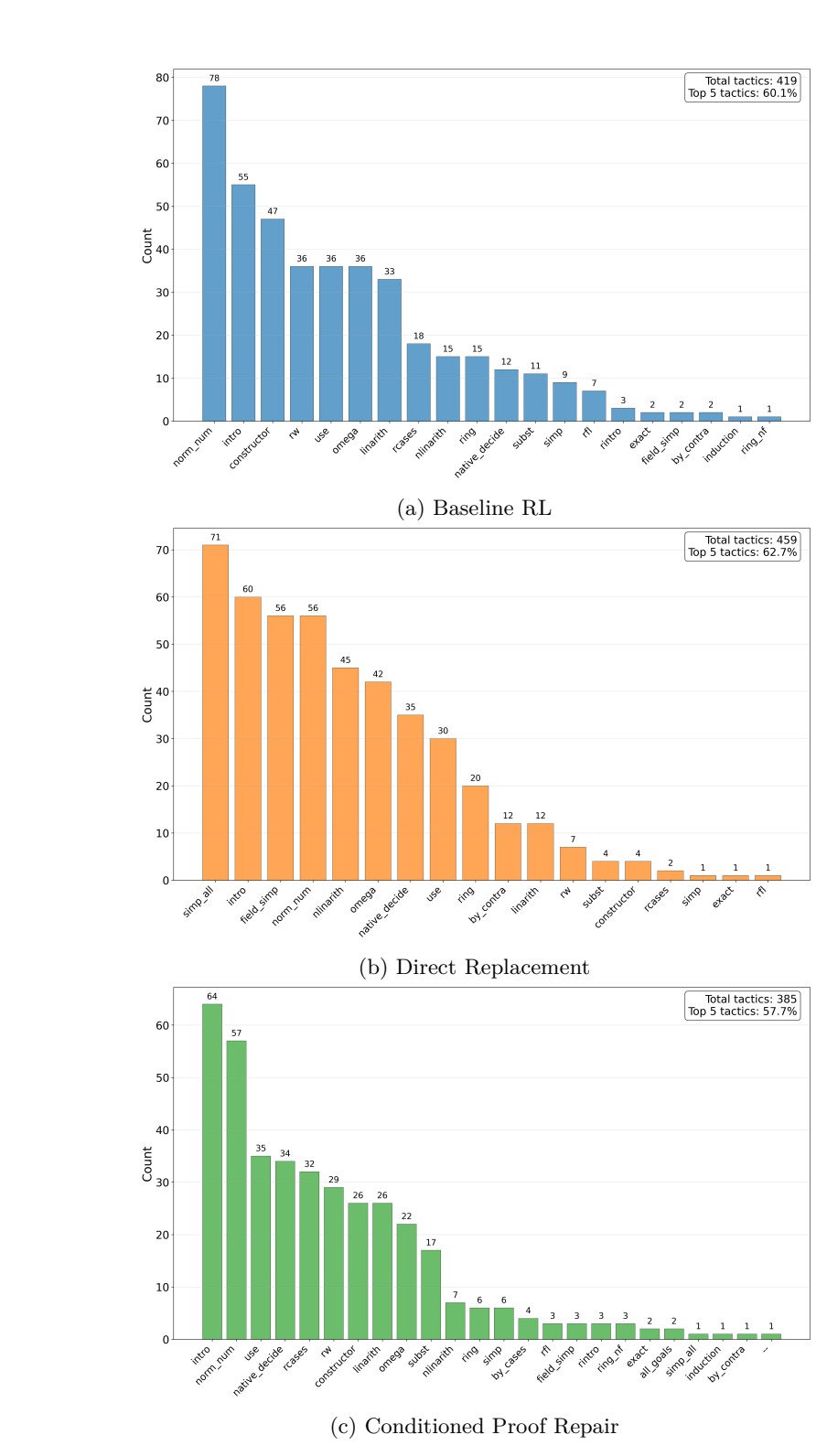

(a) Baseline RL

(b) Direct Replacement

(c) Conditioned Proof Repair

Figure 6: Tactic distribution comparison after 10 training steps

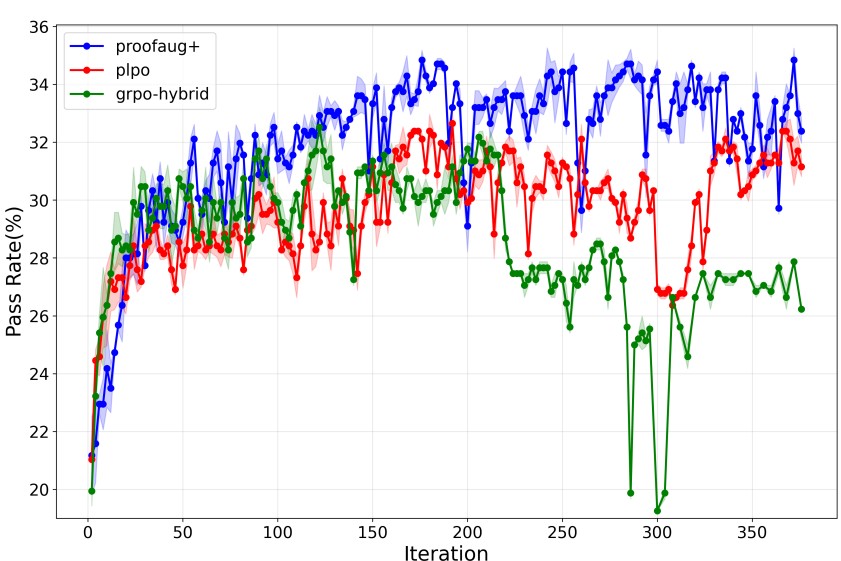

Figure 7: Performance on miniF2F.

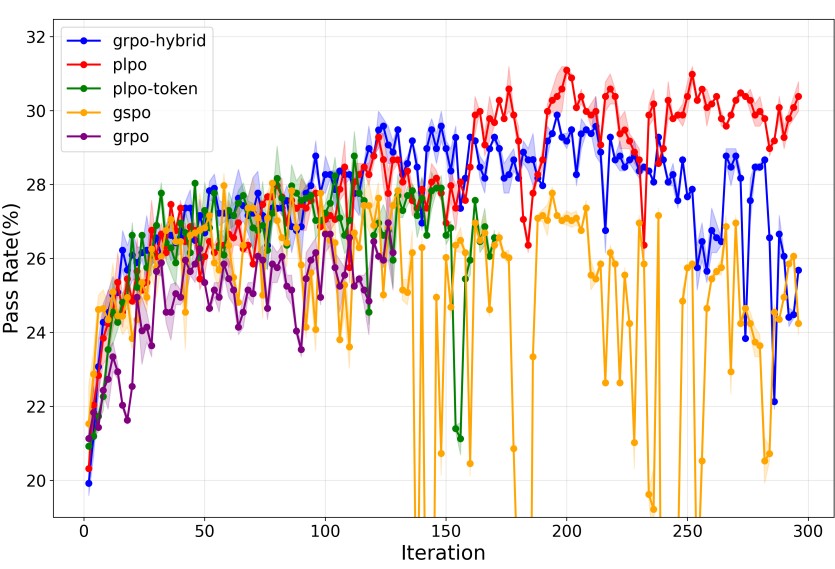

Figure 8: Comparison of PLPO, PLPO-token, GRPO and GSPO

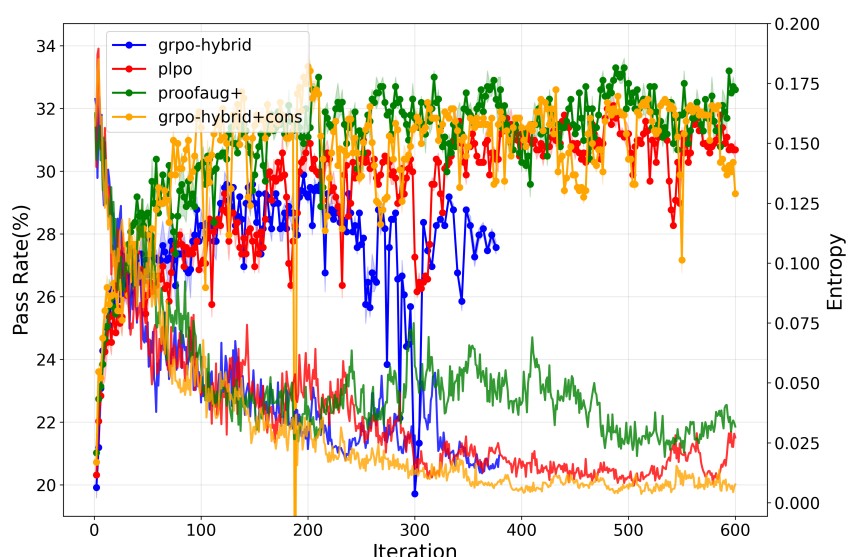

Figure 9: GRPO-hybrid with conditioned proof repair shows worse stability.

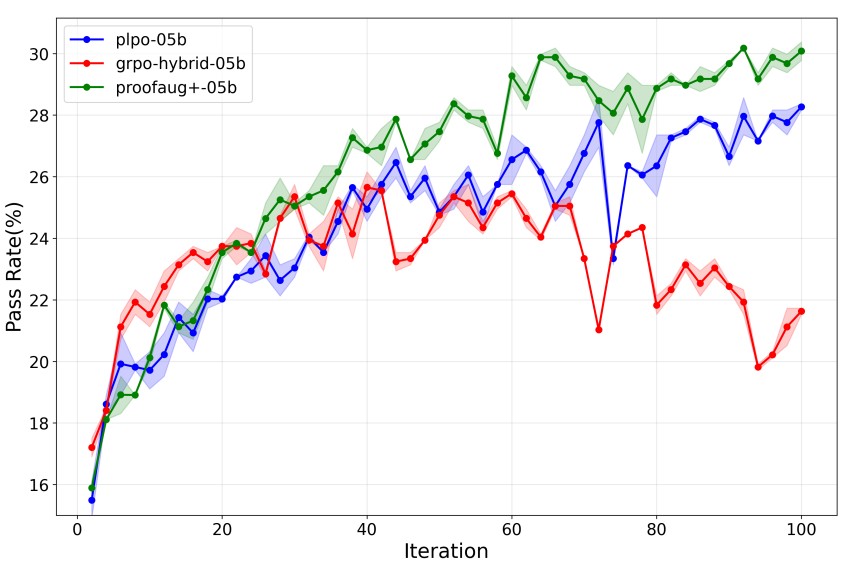

Figure 10: PLPO and ProofAug+ based on Qwen2.5-0.5B-Instruct.

