# OpenReview forum: "ProofAug+: Boosting Reinforcement Learning for LLM Theorem Provers with Conditioned Proof Repair"
_ICLR.cc/2026/Conference — Submitted to ICLR 2026_

### Official Review · Reviewer_CAUf · 2025-10-26

**Soundness:** 1
**Presentation:** 1
**Contribution:** 1
**Rating:** 2
**Confidence:** 3

**Summary:**

This paper proposes ProofAug+, a reinforcement learning training pipeline for LLM theorem provers that integrates ProofAug, an inference-time proof repair technique, to address the scarcity of positive samples in RLVR training. The design is guided by two principles: progress guarantee and variance reduction. The authors first propose PLPO (Proximal Language Modeling Policy Optimization), a novel sequence-level RL algorithm that uses exact policy objectives instead of surrogate objectives and employs a gradient rejection mechanism. They then integrate ProofAug into PLPO with constrained conditions to balance exploitation of additional positive rewards and suppression of distribution shift. Experiments on Goedel-Pset show that PLPO achieves better stability than GRPO-like baselines, and ProofAug+ yields approximately 4% performance gain over GRPO-hybrid.

**Strengths:**

- Clear motivation through preliminary experiments: Figure 1 effectively demonstrates that naive Direct-Replacement of failed proofs with ProofAug-repaired proofs leads to performance degradation, entropy collapse, and tactic collapse, establishing the need for a more principled integration approach.

- Practical engineering value: The work addresses a real challenge in applying inference-time techniques to RL training and commits to open-sourcing the complete pipeline and Lean server infrastructure.

**Weaknesses:**

1. Failure to diagnose the root cause and test the natural baseline: The paper demonstrates that Direct-Replacement fails (Figure 1) but never clearly articulates why. The fundamental issue is that Direct-Replacement creates a mismatch between the sampling distribution and the optimization objective. Specifically, the proof $y^i$ is sampled from policy $\pi$, but after applying ProofAug to obtain $y^{iO}$, the gradient is computed as $\nabla \log \pi(y^{iO}|x)$ where $y^{iO}$ was not sampled from $\pi$. This is analogous to importance sampling without correction weights. Within the standard RL framework, the natural solution to maintain sampling-optimization consistency is Reward Augmentation: sample $y^i \sim \pi(\cdot|x)$, compute reward $R(\text{ProofAug}(y^i))$, but calculate gradient as $\nabla \log \pi(y^i|x) \cdot [R(\text{ProofAug}(y^i)) - \text{baseline}]$. This approach preserves the sampling-optimization match, avoids distribution shift entirely, and still provides additional positive reward signals. The paper completely skips testing this baseline and directly jumps to introducing PLPO and conditioned repair strategies. Without demonstrating that Reward Augmentation fails, the necessity of the entire complex framework remains unjustified.

2. Disconnect between PLPO derivation and the ProofAug integration problem: Section 3.1 derives PLPO purely from general considerations of improving PPO/GRPO for LLMs. Lemma 1 proves that exact policy objectives can be used instead of surrogate objectives in the LLM setting (fixed horizon $T$, $\gamma=1$, $S_{t+1} = S_t \oplus A_t$), and the gradient rejection mechanism is motivated by questioning why importance ratios must serve as update rejection criteria. Neither derivation references the specific failure modes of Direct-Replacement or analyzes why sampling-optimization mismatch causes problems. The stated guiding principles (progress guarantee and variance reduction) are generic RL desiderata applicable to any policy optimization method, not principles specifically derived from analyzing the ProofAug integration challenge. Figure 2 confirms that PLPO alone outperforms GRPO-hybrid without any ProofAug integration, demonstrating that PLPO is an independent contribution. The paper artificially couples two separate advances—a novel RL algorithm and a heuristic for ProofAug integration—without establishing their necessary connection.

3. Experimental design cannot isolate ProofAug's contribution: Figure 2 compares three conditions: GRPO-hybrid (baseline), PLPO (new algorithm), and PLPO+ProofAug (new algorithm + integration). This design conflates two factors and cannot answer the critical question: does ProofAug integration itself provide benefits, or do the gains simply come from using a better RL algorithm? The missing crucial experiment is GRPO-hybrid with ProofAug integration using the same conditioned repair strategy (Equation 8). Without this comparison, we cannot determine whether the 4% improvement over GRPO-hybrid comes from PLPO's algorithmic superiority, from ProofAug integration, or from their interaction. Furthermore, the paper claims ProofAug helps by "acquiring more positive samples during rollout" but never directly validates this hypothesis—no experiments track positive sample counts over training or demonstrate that increased positive samples, rather than other factors, drive the performance gains.

4. Ad-hoc conditions lacking principled justification: The three conditions in Equation 8 appear to be the result of empirical tuning rather than derivation from the stated principles. Condition 1 ($d(y^{iO}) \neq 1$) rejects single-tactic proofs to "avoid learning only high-level automated tactics," but this presumes single-tactic solutions are undesirable without evidence. Condition 2 ($d(y^{iO}) \geq d(y^i)$) prevents depth decrease, but why depth must increase is not theoretically justified. Condition 3 ($\forall j, R(y^j) = 0$) only applies ProofAug when all attempts fail, which is conservative but lacks theoretical grounding. The paper provides no ablation studies removing individual conditions to establish their necessity. These conditions are highly specific to ProofAug's structural properties (maintaining proof trees with measurable depth) and would not apply to most other inference-time techniques. The paper claims ProofAug+ "can inspire integration of techniques that share similar properties" (Abstract) but never characterizes what those properties are or provides a framework for deriving analogous conditions for other methods.

5. Poor generalization contradicting the claimed benefits: The miniF2F results (Figure 7) show that ProofAug+ provides no benefit over PLPO alone on out-of-distribution data and exhibits a concerning downward trend. If the model genuinely learns better proof strategies through exposure to additional positive samples from ProofAug, it should generalize better to new problems. The observed lack of improvement suggests the model may be overfitting to ProofAug's specific repair patterns rather than acquiring transferable proving skills. The paper dismisses this negative result by suggesting the need for "more diverse and high-quality open-source formal statement dataset," but miniF2F is an established cross-system benchmark spanning multiple mathematical domains. This poor OOD performance undermines the core claim that ProofAug integration improves sample efficiency and proof-finding capabilities.

6. Limited applicability to modern reasoning models and fundamental incompatibility with Chain-of-Thought: Section 4.1 explicitly states that training data has "thinking process removed" and uses "non-CoT template." While this design choice sidesteps immediate issues, it reveals a deeper problem. Modern reasoning models like DeepSeek-Prover V2 and Kimina-Prover generate proofs through Chain-of-Thought, producing outputs like "[CoT reasoning: Let's use induction on n. The base case is trivial. For the inductive step, we need to show...] → [Proof: induction n; base case: omega; inductive case: ...]". However, if this proof attempt fails and ProofAug repairs it by replacing the entire induction strategy with a different tactic sequence like "by simp [lemma_xyz]", the result becomes "[CoT: Let's use induction... inductive step...] → [Proof: by simp [lemma_xyz]]", creating a fundamental inconsistency where the detailed strategic reasoning in CoT is completely contradicted by the actual proof. Training on such examples would teach the model that reasoning and execution can diverge arbitrarily, undermining the core purpose of Chain-of-Thought. This is not merely a limitation that could be addressed in future work—it suggests that the entire paradigm of replacement with repairing proof traces while preserving earlier reasoning is incompatible with how state-of-the-art reasoning models operate. The paper's restriction to non-CoT settings is not just a simplifying assumption but a fundamental constraint that severely limits the method's relevance to current best practices in mathematical reasoning.

7. Unclear presentation of key concepts: The term "grpo-hybrid" appears in Figure 1 without definition. Only later in Section 4.1 do readers learn it combines GRPO's group normalization, PPO's ratio clipping, DAPO's zero-KL penalty, and Dr.GRPO's LOO estimation. This baseline should be explicitly introduced before its first use with justification for why this particular combination represents a reasonable starting point.

**Questions:**

The main questions and concerns are detailed in the Weaknesses section above.

---

> ### Author Response · Authors · 2025-12-04
> **Thank you**
>
> We thank the reviewer for the insightful and constructive comments. Below are our response to your specific concerns.
>
> **W1:** Failure to clearly articulate the root cause and test the reward augmentation baseline.
>
> Since the Reward Augmentation approach gives a high reward for the wrong original traces, it will teach the model to learn to generate traces that rely on ProofAug to transform them into correct ones, which is different from the goal of training a theorem prover that can **directly generate a correct proof**. In the very prelimnary stage of our research, we indeed have tried training with this reward augmentation technique find that it leads the model to collapse completely —— it will learn to generate a single tactic for any prompt statement.
>
>
> In fact, at the beginning of section . The two principles
>
> **W2:** Disconnect between PLPO derivation and the ProofAug integration problem.
>
> A: Thank you for the insightful comment. We admit that these two parts should be stated separately and plan to reorganize the paper.
>
> **W3:** Experimental design cannot isolate ProofAug's contribution
>
> A: We have completed the experiment on GRPO-hybrid + conditioned proof repair in Figure 9. It justifies the function of ProofAug.
>
> **W4:** Ad-hoc conditions lacking principled justification.
>
> A: Thank you for pointing this out. We will address in the future.
>
>
> **W6:** Limited applicability to modern reasoning models and fundamental incompatibility with Chain-of-Thought
>
> A: Thank you for the insightful comment. In fact, ProofAug+ proposes to sample with $\pi_O$ and set restrictions on applying ProofAug. This practice   For This limitation In future, we plan to introduce a  mechanism , XXXXXXX. The all-0-reward condition naturally transfers to this Long-CoT scenario, depth-based conditions and In fact some we have tried TODO
>
> **W7:** Unclear presentation of key concepts: The term "grpo-hybrid" appears in Figure 1 without definition. Only later in Section 4.1 do readers learn it combines GRPO's group normalization, PPO's ratio clipping, DAPO's zero-KL penalty, and Dr.GRPO's LOO estimation. This baseline should be explicitly introduced before its first use with justification for why this particular combination represents a reasonable starting point.
>
> A: We agree with the reviewer. In the revised version, we have explicitly introduced GRPO-hybrid in the first use. It is reasonable in that Goedel-Prover-V2 takes the similar approach.

---

### Official Review · Reviewer_ryvb · 2025-10-31

**Soundness:** 3
**Presentation:** 3
**Contribution:** 3
**Rating:** 8
**Confidence:** 2

**Summary:**

The authors proposed a new Reinforcement Learning with Verifiable Rewards (RLVR) scheme called ProofAug+ whose design is based on two principles: progress guarantee and variance reduction. Proximal Language Modeling Proximal Optimization (PLPO), is proposed as the base algorithm of ProofAug+. The authors claim that PLPO is superior in stability and performance than popular PPO variant counterparts.

**Strengths:**

- The paper is well written and mathematically precise. Targets and gradients are spelled out with lemmas, which did help the audience better understand the paper.
- The development of the new algorithms comes with clear motivation. One can see how practical need drives the design in Section 3.2.

Disclaimer: I am not an expert in the theory of reinforcement learning. It is possible that there are mistakes in the derivation that I overlooked or didn’t understand.

**Weaknesses:**

- The empirical validation seems limited. But this is rather minor as the paper is mainly theoretical.

**Questions:**

Have you thought about how ProofAug+ may be adapted so that it works for both whole-proof generation and hierarchical proof generation?

---

### Official Review · Reviewer_9XsZ · 2025-11-01

**Soundness:** 3
**Presentation:** 3
**Contribution:** 3
**Rating:** 6
**Confidence:** 4

**Summary:**

This paper proposes ProofAug+, a reinforcement learning training pipeline for LLM theorem provers that addresses the scarcity of positive samples in formal theorem proving. The work makes two main contributions: (1) PLPO (Proximal Language Modeling Policy Optimization), a novel sequence-level RLVR algorithm that uses exact policy advantage objectives and gradient rejection mechanisms instead of traditional ratio clipping, and (2) a conditioned proof repair strategy that integrates ProofAug (an inference-time proof repair technique) into the training loop under specific constraints to balance additional positive signals against distribution shift.

The authors identify that naively replacing failed proofs with ProofAug-repaired proofs during training causes performance degradation and "tactic collapse." Guided by two principles, progress guarantee and variance reduction, they develop PLPO and carefully constrain when ProofAug is applied (only when all samples fail and the repaired proof maintains sufficient depth). Experiments on Qwen2.5-1.5B-Instruct trained on Goedel-Pset show approximately 4% improvement over GRPO-hybrid baseline, though generalization to miniF2F remains limited.

**Strengths:**

The problem formulation is well-motivated. The preliminary experiment (Section 2.2) effectively demonstrates that naive integration of ProofAug hurts performance, establishing clear motivation for the proposed approach.

The derivation of PLPO from first principles (Lemma 1) and the gradient rejection mechanism provide theoretical grounding beyond purely empirical approaches.

Decoupling gradient estimation from large update suppression via gradient rejection (rather than ratio clipping) is an interesting design choice that could have broader applications.

The paper includes ablation studies (direct replacement vs. conditioned), entropy analysis, tactic usage statistics, and multiple algorithm comparisons.

The authors provide extensive implementation details in appendices and commit to open-sourcing, which is valuable for the community.

The authors honestly discuss limitations including lack of convergence analysis, limited experimental scope, and poor miniF2F generalization.

**Weaknesses:**

Major:

1. Limited experimental validation:

1.1 Only one model size (1.5B) tested, unclear if results hold for larger models.

1.2 Only one dataset (Goedel-Pset) indicate generalization maybe be uncertain.

1.3 Poor performance on miniF2F suggests limited out-of-distribution generalization.


2. Incomplete theoretical analysis:

2.1 No formal proof of "progress guarantee", only informal arguments.

2.2 No convergence guarantees for PLPO.

2.3 The connection between the two guiding principles and the specific algorithm design is somewhat loose.

2.4 Missing analysis of why conditioned proof repair specifically addresses the identified issues


3. Ad-hoc design choices:

The conditions in Equation 8 (depth ≥ 1, depth-preserving, all-samples-fail) seem empirically motivated rather than principled. Why these specific conditions? What about other possible constraints? Limited ablation on these design choices.


4. Modest improvements:

Given the added complexity, the cost-benefit trade-off is unclear. Training efficiency gains is not quantified.



Minor:

5. Writing clarity: The "progress guarantee" and "variance reduction" principles could be more formally defined.

6. Missing comparisons: No comparison with other methods for addressing positive sample scarcity (e.g., synthetic data generation, curriculum learning). Limited discussion of alternative approaches to proof repair integration.

7. Hyperparameter sensitivity: Limited exploration of sensitivity to $ε_l, ε_h$, temperature, sample size. The choice of sum-type vs. average-type rejection not thoroughly analyzed.

8. Generalization concerns: The miniF2F results (Appendix D) show PLPO works but ProofAug+ does not help for out-of-distribution generalization. This significantly limits the practical applicability. This suggests the method may be overfitting to Goedel-Pset characteristics.

**Questions:**

1. Theoretical justification: Could you please provide a formal theorem or at least a more rigorous argument for the "progress guarantee"? Under what conditions does PLPO guarantee improvement in the objective?

2. Design choices for conditions:

2.1 How sensitive is performance to the specific conditions in Equation 8?

2.2 What happens if you only enforce the all-samples-fail condition without depth constraints?

2.3 Have you explored other possible conditioning strategies?

3. Comparison with state-of-the-art: How does ProofAug+ compare with recent work like DeepSeek-Prover-V2 or the original Goedel-Prover? What is the computational overhead compared to baseline GRPO?

4. Generalization:

4.1 Why does ProofAug+ fail to improve miniF2F performance while PLPO alone works?

4.2 Does this suggest fundamental limitations of the conditioned repair strategy?

4.3 What modifications would be needed for better out-of-distribution generalization?

5. Scalability:

5.1 Have you tested on larger models (7B, 13B+)? Do the benefits of ProofAug+ increase or decrease with model size?

5.2 What about different base model families?

6. Gradient rejection:

6.1 How does the rejection rate evolve during training?

6.2 What percentage of gradients are typically rejected?

6.3 Is there a risk of rejecting too many useful updates?

7. ProofAug application frequency: In practice, how often are the conditions in Equation 8 satisfied? What percentage of training samples actually use ProofAug repairs? Could the low application frequency explain limited benefits?

8. Alternative baselines: How does ProofAug+ compare to simpler alternatives such as:
- Just increasing the sample size for baseline GRPO?
- Using ProofAug to generate offline data for supervised fine-tuning?
- Reward shaping based on proof structure?

---

> ### Author Response · Authors · 2025-12-04
> **(Partial) Response to the questions.**
>
> We thank the reviewer for the detailed and insightful comments, and would like to first apologize for our late response due to some unexpected accidents. We also want to apologize that the response is incomplete and could not fully address your concerns. Below is our response to the questions:
>
> **Q1:** Theoretical justification: Could you please provide a formal theorem or at least a more rigorous argument for the "progress guarantee"? Under what conditions does PLPO guarantee improvement in the objective?
>
> A: We have now added a comment after Lemma 1 to claim that, under an infinitesimal learning rate, the expected total return is expected to increase for a gradient step when the number of training samples approaches infinite and makes a good estimation of this expected gradient. It can be turned into a formal theorem by making more thorough discussion.
>
> **Q2:** Design choices for conditions:
>
> **Q2.1 & 2.2:** How sensitive is performance to the specific conditions in Equation 8? What happens if you only enforce the all-samples-fail condition without depth constraints?
>
> A: We did ablataion study and found that the all-zero-reward contributes most, and the depth-based condtions could be detrimental sometimes. Nevertheless, in our exploration experiments, depth-based conditions alone work. we will further look into why.
>
> **Q2.3:** Have you explored other possible conditioning strategies?
>
> A: Empirically, I have tried directly adding a depth multiplied by an coefficient as reward and found it detrimental. And also execution time.
>
> **Q3:** Comparison with state-of-the-art: How does ProofAug+ compare with recent work like DeepSeek-Prover-V2 or the original Goedel-Prover? What is the computational overhead compared to baseline GRPO?
>
> A: These models achieve >50\% performance on miniF2F, which is not comparable for ProofAug+, that only achieves 35\% performance on miniF2F for 1.5B models. The computational overhead is near GRPO since we have optimized the implementation of the all-samples-fail mechanism to be only one-pass. Also, we work in an 1-step async setting and the Lean-server signal can be returned before next step RL.
>
> **Q4:** Generalization: Why does ProofAug+ fail to improve miniF2F performance while PLPO alone works?
> A: We are sorry that the poor performance on miniF2F was resulted by our miss in implementing PLPO. As you can see in the original supplementary material, we mistakenly forget to mask the prompt part when calculating the gradient and probablity ratio, which might have caused biased estimation and gradient rejection. After fixing the implementation, performance on miniF2F shows consistent result with that on Goedel-Pset test split.
>
> **Q5:** Scalability:
>
> **Q5.1:** Have you tested on larger models (7B, 13B+)? Do the benefits of ProofAug+ increase or decrease with model size?
>
> A: We have tested Qwen2.5-0.5B-Instruct for the same pipeline. According to Figure 10, it shows consistent performance. We hope that in the future we could test on larger models. It turns out that the benefit of ProofAug+ over PLPO maintains similar in this 0.5B setting.
>
> **Q5.2:** What about different base model families?
>
> A: We will consider complete this ablation in the future. We expect for Intruction models (rather than thinking model) the performance should be the same.
>
> **Q6:** Gradient rejection: How does the rejection rate evolve during training? What percentage of gradients are typically rejected? Is there a risk of rejecting too many useful updates?
>
> A: For GRPO-hybrid, it decrease from ~5e-4 to ~2e-4 during the training. For PLPO, it is ~0.05 to ~0.01. For ProofAug+, it is from ~0.06 to ~0.03. We think this fraction is not too much.
>
> **Q7:** ProofAug application frequency: In practice, how often are the conditions in Equation 8 satisfied? What percentage of training samples actually use ProofAug repairs? Could the low application frequency explain limited benefits?
>
> A: We record the value of original trace rewards, full ProofAug rewards and conditioned ProofAug rewards (true training rewards of ProofAug+) during the training. In step 1, it is 0.1543, 0.3047 and 0.1602; In step 100, it is 0.2441, 0.2969 and 0.2461. We think the low application frequency indeed explain the limited benefits.
>
> **Q8:** Alternative baselines: How does ProofAug+ compare to simpler alternatives such as:
> - Just increasing the sample size for baseline GRPO?
>     - A: We think this is a good practice, but will be slower since the sample generation takes up most of the time in RL.
> - Using ProofAug to generate offline data for supervised fine-tuning?
>     - A: We have tried this baseline and found it slower than RL training.
> - Reward shaping based on proof structure?
>     - A: At least we have ripped out the possibility of depth-based reward for reward shaping.
>
> Again, we thank the reviewer for the great effort devoted to the comments, and deeply apologize for not having magnaged to write a full response.

---

### Official Review · Reviewer_a6NQ · 2025-11-03

**Soundness:** 3
**Presentation:** 2
**Contribution:** 3
**Rating:** 4
**Confidence:** 3

**Summary:**

This work proposes an improvement in RLVR for theorem proving, by incorporating ProofAug, a previously known inference-time algorithm that repairs traces that are unsuccessful. They also introduce a new algorithm, PLPO, which is a modified version of GRPO (with RLOO for the baseline).

This paper first tries a naive version of ProofAug+, where in the gradient updates, the responses are directly replaced with the ProofAug corrected versions. This worsens the accuracy significantly, compared to vanilla GRPO. Thus, they introduce a new method PLPO, which modifies RLOO with the goal of training on the Proof Aug-generated proofs, which are off-policy. The PLPO gradient decouples importance sampling from the criteria used to reject the gradient. Additionally, instead of calculating token-level advantages as in GRPO, they switch to calculating sequence-level advantages as in GSPO. They argue that a downside of RLOO is that the baseline is not related to the prefix in the case of token-level advantages.

Another major contribution is introducing the corrected proofs generated by Proof Aug into the RL algorithm. In the naive algorithm described in section 2.2, in the gradient update, the original proof is directly replaced with the proof generated by Proof Aug. Instead, in the more principled version proposed in Section 3.2, the importance sampling ratio is calculated correctly, using \pi_O as the denominator (where \pi_O(y) is the probability that a proof y can be generated by first rolling out using \pi, then modifying it with Proof Aug). Proofs generated by Proof Aug are incorporated only when necessary, i.e. the original proofs all fail, and Proof Aug is not used if it reduces the depth of the proof.

In the experiments, this work uses Qwen2.5-1.5B-Instruct, and trains on Goedel-Pset, first using some cold-start data from Kimina-Prover-1.7B. The model is evaluated on a separate subset of Goedel-Pset. They find that PLPO improves the performance of GRPO-hybrid (from the OpenRLHF codebase) and incorporating Proof Aug further improves the performance significantly.

**Strengths:**

1. The improvement using PLPO is significant, and the performance further improves when incorporating Proof Aug.
2. It is interesting that incorporating Proof Aug can help, since it may be quite off-policy.

**Weaknesses:**

1. The writing clarity can perhaps be improved.
    1. For example, in the section 2.2 dealing with the naive approach, formally state in an equation what this version of the approach is doing? (I think in the later section about PLPO and incorporating ProofAug, it is alluded to. But it would be helpful to explicitly state it in Section 2.2)
    2. What is the sum-type rejection criteria in Equation (6)? It sounds pretty strange that this might work on the sequence level, since the ratio will likely be either very large or very small. It seems like $\varepsilon_l$ and $\varepsilon_h$ would have to be quite big.
    3. Additionally, could you explain what GRPO-hybrid is and how it may differ from the usual GRPO (or otherwise just state that it is GRPO)?
    4. It is also pretty surprising that in Equation (16), you can use the ratio of $\tilde\pi$ and $\pi_O$ for importance sampling, without any sort of normalization. Am I missing something here?
    5. It would be useful to add some verbal explanation of ProofAug in the appendix (currently there is only a figure which is taken from the ProofAug paper).
    6. Figure 1: It might make more sense to have the entropy be in a different figure? Also, add titles to all of the figures.

**Questions:**

1. Typos
    1. Line 155: “ProofAug repairs in” -> “ProofAug repairs it”
    2. Line 161: “We also make a statistics -> We also collect statistics”
2. At the beginning of section 3, could you explain why “progress guarantee” and “variance reduction” are the issues with the naive version that is tried in Section 2.2?
3. Could you clarify why you use $\varepsilon_l=0.2$ and $\varepsilon_h=0.27$ for GSPO? The original GSPO paper uses much lower values of $\varepsilon_l$ and $\varepsilon_h$.
4. In Equation (16), how would you set $\varepsilon_l$ and $\varepsilon_h$ for gradient rejection? I would imagine that $\pi_O$ and $\tilde\pi$ are quite different, so their ratio might not be even close to 1 at the beginning.
5. It seems ProofAug+ also leads to an entropy collapse in Figure 2, curious why that is.

---

> ### Author Response · Authors · 2025-12-04
> **Thank you for the comments. Response to Weaknesses**
>
> We thank the reviewer for the efforts on contributing the insightful comments. Below are our response to your concerns.
>
> ### Response to Weaknesses
>
> **W1:** The writing clarity can perhaps be improved.
>
> **W1.1 & W1.3:** In the section 2.2 dealing with the naive approach, formally state in an equation what this version of the approach is doing? Additionally, could you explain what GRPO-hybrid is and how it may differ from the usual GRPO?
>
> A: Thank you for the advice. We have added a formal equation to state what GRPO-hybrid is, and state that the direct-replacement is to replace all the response traces with the ProofAug version.
>
> **W1.2:** What is the sum-type rejection criteria in Equation (6)? It sounds pretty strange that this might work on the sequence level, since the ratio will likely be either very large or very small. It seems like  $εl$  and  $εh$  would have to be quite big.
>
> A: Thank you for the comment. The sum-type rejection requires that the ratio of the trace probability under the target policy over the that under the reference policy should be controlled according to the estimated advantage. While in section 3.1 we have shown an ideal experiment that assumes all token-wise ratios subject to the Gaussian distribution could make the sum-type ratio explode, the explosion rate is around $\sqrt{n}$, which is moderate when the response length $n$ is not that large. In our experiments, the $\epsilon_l=0.2$ and $\epsilon_h=0.28$ choice maintains a rejection rate of lower than 5% during the training. This justifies the practicle use of the sum-type rejection criteria.
>
> **W1.4:** It is also pretty surprising that in Equation (16), you can use the ratio of  $π~$  and  $πO$  for importance sampling, without any sort of normalization. Am I missing something here?
>
> A: We feel sorry for the confusion. The original Equation (16) is the target that we would like to approximate. We in fact reduce the gap between $\tilde{\pi}/\pi_O$ and $\tilde{\pi}/\pi$ through the conditions. Now we have modified the presentation in section 3.2 to emphasize that the current Equation (17) is the ideal target rather than what we obtain in practice. We also guess that you are worry about there should be a large gap between $π~$ and  $π_O$, which could elicit gradient rejection, thus prohibiting the model from learning from the positive traces. We argue that for the first time in each iteration the target policy encounters a trace it is not familiar with, it will probably learn it since now it is not far away from the reference policy (and only in the subsequent steps where it faces similar traces, gradient rejection mechanism is invoked). As a result, such practice strikes a balance between the learnability and stability.
>
> **W1.5:** It would be useful to add some verbal explanation of ProofAug in the appendix (currently there is only a figure which is taken from the ProofAug paper).
>
> A: Thank you for the suggestion. We have added an extra verbal explanation of ProofAug in the appendix in the revised paper.
>
> **W1.6:** Figure 1: It might make more sense to have the entropy be in a different figure? Also, add titles to all of the figures.
>
> A: This is a nice suggestion and we believe this can make the presentation clearer. However, due to the page limit, there is hardly any spece for another subfigure for entorpy. As a compensation, we have adjusted Figure 1 in order that the entropy curve becomes clearer and add descriptions on the curve meanings in the caption.

---

> > ### Author Response · Authors · 2025-12-04
> > **### Response to Questions**
> >
> > **Q1:** Typos. Line 155: “ProofAug repairs in” -> “ProofAug repairs it”. Line 161: “We also make a statistics -> We also collect statistics”.
> >
> > A: Thank you for pointing them out. We have fixed the typos in the revised version.
> >
> > **Q2:** At the beginning of section 3, could you explain why “progress guarantee” and “variance reduction” are the issues with the naive version that is tried in Section 2.2?
> >
> > A: Thank you for the advice. We have added some statements on how the naive version break the two issues.
> >
> > **Q3:** Could you clarify why you use  $ε_l=0.2$  and  $ε_h=0.27$  for GSPO? The original GSPO paper uses much lower values of  $ε_l$  and  $ε_h$ .
> >
> > A: We have make double-check that the value used in GSPO is indeed $εl=0.2$ and $εh=0.27$. In their section 5.1, they say "We compare against GRPO as the baseline and set the left and right clipping ranges in Equation (2) (GSPO expression) to 0.2 and 0.27".
> >
> > **Q4:** In Equation (16), how would you set  $ε_l$  and  $ε_h$  for gradient rejection? I would imagine that  $πO$  and  $π~$  are quite different, so their ratio might not be even close to 1 at the beginning.
> >
> > A: Since we directly use $\pi$ to estimate $\pi_O$, at the beginning the estimated ratio will be 1 and allows update. Section 3.2 states that we add some restrictions on applying ProofAug, thus the frequency of such biased estimations are controlled, reducing the extent that the training is affected by this bias. Later in the training, the model alreadly learns a lot from ProofAug, and the gap betweeen the estimated and the real ratio become closer.
> >
> > **Q5:** It seems ProofAug+ also leads to an entropy collapse in Figure 2, curious why that is.
> >
> > A: Sorry for the confusion. The story is that, in Figure 1, we observe that direct-replacement accelerates the entropy collapse. So we are just finding a method to incorporate ProofAug that will not accelerate the collapse, as shown in Figure. To get better pass@1 performance, entropy collapse is needed by the end. The pass@k training papers (arxiv:2508.10751 and arXiv:2505.15201) illustrate this question well. Our goal is just not to accelerate the collapse.
> >
> > Again, we thank the reviewer for the great efforts on reviewing, and hope our response have addressed your concerns.

---

### Author Response · Authors · 2025-12-04
**Thank all the reviewers and apologize**

We thank all the reviewers for their constructive and insightful comments. We feel sorry that we fail to address all the concerns of the reviewers due to lack of time for rebuttal resulted by unexpected accidents. We would like to update a partially revised version and partial response to answer the questions of the reviewers and withdraw, since we also feel that the paper indeed need reorganize and justify more on its claims.

---

### Meta-Review · Area_Chair_guXm · 2026-01-07

**Summary:**

This paper attempts to improve RL for theorem proving, by incorporating ProofAug, a previously known inference-time algorithm that repairs traces that are unsuccessful. They also introduce a new algorithm, PLPO, which is a modified version of GRPO (with RLOO for the baseline), and run these together.

Many of the reviews were a bit less detailed, but I agree with the concerns of the last reviewer CAUf: lack of experimental comparisons to several other off-policy RL methods for LLMs (such as LUFFY), adhoc justification, poor quality of writing, and imperfect diagnosis of root causes behind the problems observed. The paper is on a right track and I am sorry to see that the authors could not provide a rebuttal due to an accident (I hope they are doing fine!) but unfortunately we cannot accept the paper at this stage.

**Reviewer Concerns:**

Most of the reviewer CAUf's concerns are outstanding and not addressed. Some of the other reviewers' concerns are primarily clarification based so they are addressed.

**Reviewer Scores:**

I think the Reviewer CAUf would not have changed their scores for the most part.

---

### Decision · Program_Chairs · 2026-01-26

Reject